# Angular-Based Radiometric Slope Correction for Sentinel-1 on Google Earth Engine

**Andreas Vollrath** [1,*], **Adugna Mullissa** [2] and **Johannes Reiche** [2]

1 European Space Agency, ESRIN, ESA Phi-Lab, Largo Galileo Galilei, 00044 Frascati (RM), Italy
2 Department of Environmental Sciences, Wageningen University and Research, Droevendaalsesteeg 3, 6708 PB Wageningen, The Netherlands; adugna.mullissa@wur.nl (A.M.); johannes.reiche@wur.nl (J.R.)
* Correspondence: andreas.vollrath@esa.int; Tel.: +39-06-941-80222

**Abstract:** This article provides an angular-based radiometric slope correction routine for Sentinel-1 SAR imagery on the Google Earth Engine platform. Two established physical reference models are implemented. The first model is optimised for vegetation applications by assuming volume scattering on the ground. The second model is optimised for surface scattering, and therefore targeted at urban environments or analysis of soil characteristics. The framework of both models is extended to simultaneously generate masks of invalid data in active layover and shadow affected areas. A case study, using openly available and reproducible code, exemplarily demonstrates the improvement of the backscatter signal in a mountainous area of the Austrian Alps. Furthermore, suggestions for specific use cases are discussed and drawbacks of the method with respect to pixel-area based methods are highlighted. The radiometrically corrected radar backscatter products are overcoming current limitations and are compliant with recent CEOS specifications for SAR backscatter over land. This improves a wide range of potential usage scenarios of the Google Earth Engine platform in mapping various land surface parameters with Sentinel-1 on a large scale and in a rapid manner. The provision of an openly accessible Earth Engine module allows users a smooth integration of the routine into their own workflows.

**Keywords:** radiometric slope correction; Google Earth Engine; Sentinel-1; Analysis-Ready-Data

## 1. Introduction

Google Earth Engine (GEE) was one of the first web-based platforms adopting the paradigm shift of providing Earth Observation Analysis-Ready-Data (ARD) on a Big Data infrastructure for rapid large-scale analytics of geo-spatial datasets [1]. The provision of ARD supersedes the cumbersome and computationally intensive burden of preprocessing low-level satellite data for the user. It therefore allows for immediate access of the imagery and let the user focus on the actual information extraction. This is especially attractive for working with Sentinel-1 imagery, as the complexity of preprocessing Synthetic Aperture Radar (SAR) data is one of the main reasons for its slow uptake by a wider user community [2].

At the moment, there is no unique consensus on the specification of ARD products for SAR backscatter. Before the ingestion of the Sentinel-1 data onto the GEE platform, basic processing steps (noise removals, calibration and geocoding) are applied [3], resulting in geometrically terrain-corrected ARD products. However, to comply with the ARD data standards for land, suggested by the Committee on Earth Observation Satellites (CEOS) [4], the radiometric slope correction and the provision of an invalid data mask over areas affected by layover and shadow are required as well.

Radiometric distortions over rugged terrain within the backscatter products on GEE originate from the side-looking SAR imaging geometry and are strong enough to exceed weaker differences of

the signal due to variation in land cover [5]. It is therefore important to account for these effects during the generation of higher level backscatter products in order to enable a variety of land applications such as the robust retrieval of bio-geophysical parameters (e.g., biomass, soil moisture), land use/land cover classifications as well as complex combinations of overlapping imagery from ascending and descending orbits or different sensors [6]. However, any slope correction procedure inevitably relies on the underlying terrain geometry provided by the auxiliary information in form of a Digital Elevation Model (DEM) that ideally needs to match the geometrical resolution. In practice, this requirement is complicated to meet, especially as no DEM is freely available in the resolution of Sentinel-1, which covers the full globe. Additionally, artefacts are likely to be introduced due to land cover changes between the acquisition date of the DEM and the scene itself. Land use practices such as deforestation will affect the height model and therefore any correction routine. Consequently, it is comprehensible that the developers of GEE decided to not include this processing step in order to prevent the introduction of erroneous data [3]. Although this features the advantage of having free choice on the use of the DEM and the model applied, it leaves the implementation to the user, which in turn necessitates a fundamental understanding of the SAR acquisition principles.

An important aspect for a customised implementation of a radiometric slope correction on GEE is that the Sentinel-1 imagery is already geocoded and therefore in map geometry. While the CEOS ARD specification refers to three methods for the slope correction [5–7], only the method described in [6] allows for directly mitigating the radiometric distortions in the map geometry of the SAR image by considering the angular relationships. The two physical reference models therein specified, operate independently of terrain type, frequency as well as polarisation, and therefore are universally applicable. While one model is optimised for volumetric scattering, and thus is targeted on vegetated surfaces, the other one assumes surface backscatter from the ground and is rather suited to areas where bare ground prevails.

As with all angular-based approaches, both models feature the drawback of not accounting for the apparent heteromorphic relation between map and radar geometry as described by [5]. Both the authors of [5,8] show that pixel-area based approaches for the radiometric slope correction are more accurate by considering the actual topological relationships between both geometries and thus take all the underlying basic properties of the radar image acquisition into account. However, the back-and-forward computation between map and radar geometry of each image would not only heavily affect performance, but also requires the availability of the state orbit vectors (i.e., the exact position of the satellite during the acquisition), an information that is left out during the ingestion process on GEE. As a result, the selection of an adequate correction procedure on the GEE platform is limited and the use of an angular-based approach that is based on simplified assumptions remains the only feasible option under the current preconditions.

The effects of layover and shadow are as well linked to the side-looking imaging geometry of a SAR system and result from irreversible geometric distortions over mountainous areas. As the modelling framework of the radiometric slope correction from the work in [6] is based on angular relationships between the sensor and the terrain, it is straightforward to map areas of active layover and shadow following the method described in [9]. Again, this method is based on the simplified assumption of angular dependencies and does not take into account the topological relations between neighbouring cells that actually need to be considered to fully describe those effects. For this reason, and with respect to other methods [7,10,11], the chosen approach also lacks the mapping of passive layover and shadow areas, but has the advantage of being directly applicable in the map geometry, thus having little impact on the overall performance.

This paper provides the implementation of the two angular-based reference models (volume and surface scattering) for the radiometric slope correction and the masking of active layover and shadow areas on GEE. In Section 2, we first give a brief review of the methods and their implementation on GEE. On the basis of a case study over the Austrian Alps we demonstrate the improvements for a single Sentinel-1 scene in Section 3. A guidance on usage scenarios as well as limitations and performance are

discussed in Section 5. Section 6 summarises the main outcome of the study. In addition, the method is provided as an Earth Engine module and reproducible code used for the case study is shared via Jupyter notebooks (Supplementary Materials).

## 2. Method

### 2.1. Sentinel-1 Data Ingestion into Google Earth Engine

Imagery within the Sentinel-1 image collection of Google Earth Engine [3] is based on Sentinel-1 Ground Range Detected (GRD) products. Within this collection, all products have been already preprocessed using the European Space Agency's (ESA) Sentinel-1 Toolbox (S1TBX) [12] by applying the following processing steps.

- Apply Orbit file
- Remove thermal noise
- Remove GRD border noise
- Radiometric calibration to $\sigma^0$
- Range-Doppler terrain correction

The final output are the geocoded backscatter bands calibrated to the normalized radar cross section $\sigma^0$ in dB scale. In addition, a band containing the nominal incidence angle $\theta_i$ and basic product metadata are added [3].

### 2.2. Radiometric Slope Correction

For the Sentinel-1 based radiometric slope correction in GEE we implement the two established physical reference models presented in [6]. Both reference models depend on the angular relation- ships between the SAR image and the terrain geometry as schematically shown in Figure 1. The starting point for the implementation of both models into GEE are four angles, from which a simplified relation between image and terrain can be derived. The definitions and theoretical derivations in the following subsections are taken from work in [6].

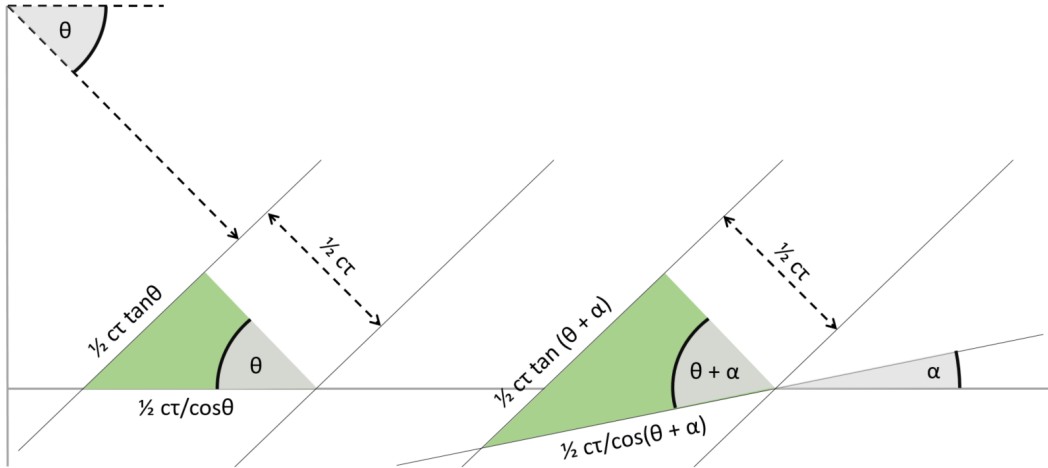

**Figure 1.** Geometry of resolution cell in range direction adapted from [13]: for flat terrain (left triangle) and facing slope (right triangle). The angle of slope steepness in range direction is $\alpha$ (or $\alpha_r$); the incidence angle $\theta_i$ equals $90° - \theta$, the range resolution is $1/2c\tau$ (or half pulse length; where pulse length is the product of speed of light $c$ and pulse duration $\tau$).

#### 2.2.1. Radar Geometry

Two angles define the radar look direction, the (nominal) incidence angle $\theta_i$ and the range (or look) direction $\phi_i$. The incidence angle $\theta_i$ is defined as the angle between the flat earth's normal direction

and backscatter direction, and increases with range distance. The standard mode over land for the Sentinel-1 mission is the Interferometric Wide Swath Mode (IW), where the incidence angle ranges between 31° and 46° from near to far range [14]. The latter is given as an auxiliary band for each Sentinel-1 image on GEE and thus directly accessible.

The range (or look) direction $\phi_i$ is the angle in the horizontal plane with respect to true North, and varies with latitude. As this information is not available in the Sentinel-1 metadata of GEE, it is approximated by calculating the direction of the gradient from the incidence angle band as proposed by the authors of [15].

### 2.2.2. Terrain Geometry

The terrain geometry is given by the slope steepness $\phi_s$ and slope aspect angle $\alpha_s$ relative to true north. The terrain geometry cannot be derived by the image itself and needs to be modelled by a Digital Elevation Model (DEM) that should ideally be in the same resolution regime (or higher) than the image itself. The aspect and slope angles are derived from the height values of a given pixel in relation to its neighbouring pixels by interpolation. Within GEE it is straightforward to calculate both angles with the *ee.Terrain* class for a given DEM.

### 2.2.3. Model Geometry

The simplified relation between image and terrain geometry is described by the slope steepness in range $\alpha_r$ and the slope aspect in azimuth $\alpha_r$. As a first step, the above mentioned four angles are reduced to three by subtracting the terrain's slope aspect angle from the SAR range direction as follows.

$$\phi_r = \phi_i - \phi_s \tag{1}$$

Subsequently, the two required angles $\alpha_r$ and $\alpha_{az}$ can be calculated:

$$\alpha_r = \arctan(\tan(\alpha_s)\cos(\phi_r)) \tag{2}$$

$$\alpha_{az} = \arctan(\tan(\alpha_s)\sin(\phi_r)) \tag{3}$$

Together with the backscatter and nominal incidence angle $\theta_i$, those angles are the basis for the subsequent reference model calculations.

### 2.2.4. Reference Models

Before applying the actual slope correction routine, the Sentinel-1 backscatter needs to be reconverted from dB into its original, linear power scale (i.e., de-logarithmised). As the data is calibrated to the normalised radar cross section $\sigma^0$ [3], the backscatter values are also affected by the incidence angle from near to far range. In order to remove this variation a correction in the form of

$$\gamma^0 = \sigma^0 / \cos(\theta_i) \tag{4}$$

needs to be computed before the relief modulation factor can be applied. It should be noted that the incidence angle $\theta_i$ on GEE is given as the viewing incidence angle, therefore neglecting the earth curvature's influence on $\theta_i$ on the ground. Thus, the resulting $\gamma^0$ of Equation (4) represents only an approximated estimate.

Finally, the relief modulation factor is expressed as the ratio of the backscatter coefficient on tilted terrain $\gamma^0$ and the backscatter on flat terrain $\gamma^0_f$. The two reference models differ in the way this relief mitigation factor is determined.

The first model (Model 1) assumes the terrain as an opaque volume of isotropic scatterers with a constant scatterer density per volume unit [13]. It is based on the ratio between the observed volume on tilted terrain (right triangle in Figure 1) in a particular range cell and the volume that would have been observed over flat terrain (left triangle in Figure 1) as follows,

$$\gamma_f^0 = \gamma^0 \frac{\tan(90 - \theta_i)}{\tan(90 - \theta_i + \alpha_r)}. \tag{5}$$

The second model (Model 2) describes the terrain as a surface of isotropic scatterers, with a constant scatterer density per tilted surface unit [16]. In this case, the tilt in azimuth direction, expressed as $cos(\alpha_{az})$ in the nominator of Equation (6), needs to be included as follows,

$$\gamma_f^0 = \gamma^0 \frac{\cos(\alpha_{az}) \cos(90 - \theta_i + \alpha_r)}{\cos(90 - \theta_i)}. \tag{6}$$

Ultimately, the backscatter data is reconverted to the dB-scale and the original metadata is copied to the properties of the corrected image.

### 2.3. Layover and Shadow Mask

SAR images are taken in a side-looking configuration, resulting in a specific image geometry called slant range (Figure 2). The formation of the image in slant range is a product of the run time between transmission and reception of the radar pulse. On flat terrain, the sequence of each pixel in the recorded slant range image follows the sequence on the actual ground (i.e., ground range). On rugged terrain, this sequence may be disturbed, as the top of a mountain might be at a further distance to the nadir of the radar antenna (i.e., distance in ground range), but not to the antenna itself (i.e., distance in slant range). This effect is called layover and appears when the slope steepness in range ($\alpha_r$) exceeds the incidence angle ($\theta_i$) on a slope facing towards the sensor (foreslope). However, the affected area is larger than just the pixels featuring this specific geometrical configuration. As shown in Figure 2a, neighbouring areas of the affected slope are within the area of the inverted isolines, splitting the layover region into subregions of active (*red line*) and passive layover (*blue lines*).

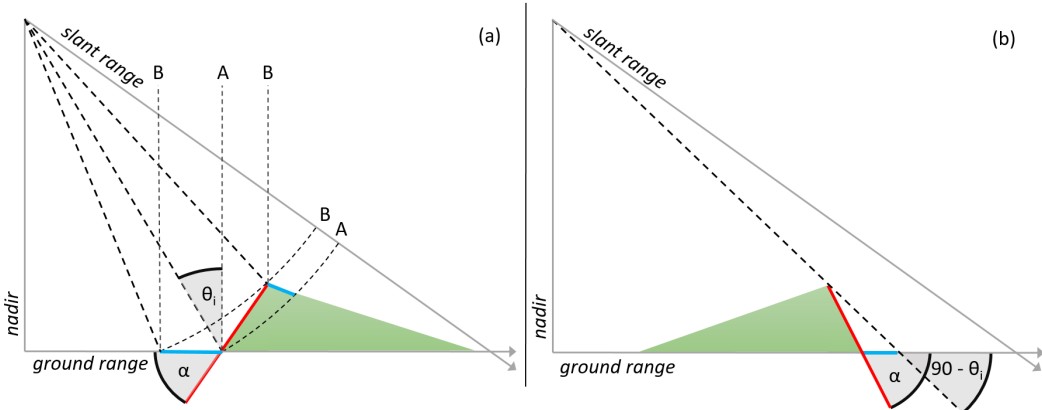

**Figure 2.** Simplified slant and ground range geometry in case of layover (**a**) and shadow (**b**). In case of layover, the backscatter of point B reaches the satellite before the backscatter of point A, which leads to a geometrical inversion in the slant range. This effect occurs when the angle $\alpha$ of the foreslope is steeper than the incidence angle. The red line depicts active layover areas that can be derived from the angular dependencies, whereas the blue lines indicate passive layover. In case of shadow, the backslope is steeper than the look angle ($90 - \theta_i$). The red line is situated on the active shadow part, while the blue line represents passive shadow.

Another effect resulting from the side-looking image geometry is called radar shadow and occurs when a slope facing away from the sensor (backslope) exceeds the look angle, defined here as $\theta = 90° - \theta_i$ (Figure 2b). In this case, the emitted radar beam does not reach the ground at all. Again, the shadow region consists of an active (*red line*) and a passive subregion (*blue line*), as neighbouring areas behind that slope are obscured by the active region as well.

As the relations between active and passive regions are complex [17], and should be ideally computed in the slant range geometry by integrating all the affected neighbouring grid cells [11], our approach of mapping layover and shadow affected areas is adopted on the simplified assumptions described by [9] as follows.

$$active\ layover = \alpha_r > \theta_i \tag{7}$$

$$active\ shadow = \alpha_r < -(90° - \theta_i) \tag{8}$$

As the slope steepness in range $\alpha_r$ is already calculated during the elaboration of the slope correction models (Equation (2)), the identification of the active layover and shadow areas is straightforward by applying the conditions of Equations (7) and (8).

The actual mapping of passive layover and shadow is not included due to the practical limitation on GEE of not being able to reconstruct the slant range. As an alternative, we introduce a customisable buffer parameter that applies a morphological filter to both masks using a circular kernel with a radius given as customisable parameter in meters. In order to keep the masking independent of GEE's scale dependent pixel size, the *fastDistanceTransform* function is used to identify neighbouring pixels of the layover and shadow affected areas independent of the zoom level. The final mask is added as a band to the corrected image of slope corrected backscatter and can be directly applied on the backscatter layers with GEE's *updateMask* function.

## 3. Case Study

### 3.1. Study Area and Data

A 30 km × 20 km study site in the high-altitude Austrian Alps, surrounding the city of Innsbruck, is selected. The elevation ranges from 500 m up to 2850 m above sea level and slope angles amount up to 85° (Figure 3a). Land cover is dominated by coniferous forests on the lower mountain slopes, herbaceous vegetation and settlement areas in the valleys as well as bare rock and scree on the slopes above the tree line (Figure 3b).

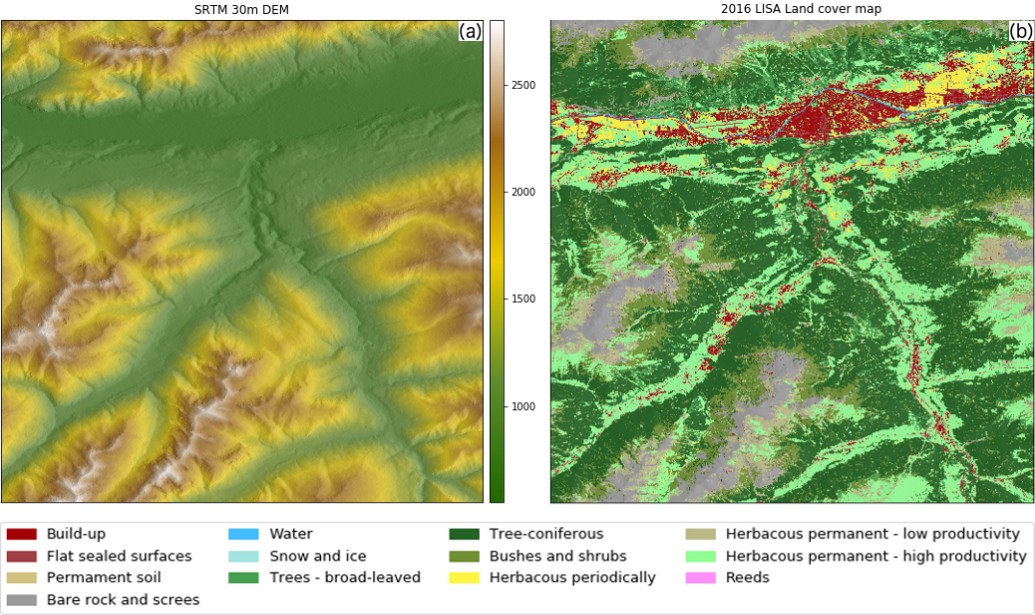

**Figure 3.** Overview of the study site with the city of Innsbruck in the central northern parts showing (**a**) the elevation based on SRTM 30 m DEM [18] and (**b**) the Level-2 Land Cover map based on [19] with the corresponding legend.

The corrections for both models are applied to a dual-polarized VV/VH Sentinel-1 image that was acquired on 15 August 2016 in the standard Interferometric Wide Swath (IW) mode. The SRTM 30 m DEM from Earth Engine's *SRTMGL1_003* collection was used for the derivation of the terrain geometry.

### 3.2. Evaluation Scheme

A first assessment of the effectiveness of the correction is done by visually inspecting the corrected image. It should be notable that differences in colour are solely related to differences in land cover and should not coincide with topography, although both can be related.

As the backscatter mechanisms change with land cover, any further statistical assessment of the improvement regarding the applied slope correction needs to consider the individual examination within a single land cover class. The national 10 m Land Cover map from Austria for the year 2016 [19] was used to divide the data points of the image for the 13 land cover classes present in the study area (Figure 3b). Due to the sparseness of data points, as well as some land cover classes, such as built-up areas, being predominantly present over flat or only slightly inclined slopes, a subset of six representative classes was selected for the evaluation (Table 1).

**Table 1.** Slope effect statistics for different land cover classes for the VV- and VH-polarisation. Mean backscatter ($\mu$), standard deviation of backscatter ($\sigma$), amplitude of backscatter as a function of slope aspect angle ($A$) and backscatter increase per degree slope steepness in range ($s$) for both models.

| | $\mu - VV$ | $\sigma$-VV | $s$-VV | $A$-VV | $\mu$-VH | $\sigma$-VH | $s$-VH | $A$-VH |
|---|---|---|---|---|---|---|---|---|
| *Trees—broad-leaved* | | | | | | | | |
| Original | −8.584 | 4.433 | 0.176 | 4.469 | −14.144 | 4.208 | 0.160 | 4.024 |
| Model I | −7.900 | 3.276 | −0.013 | 1.398 | −13.460 | 3.250 | −0.029 | 1.308 |
| Model II | −8.947 | 3.519 | 0.074 | 2.272 | −14.507 | 3.390 | 0.058 | 1.845 |
| *Tree—coniferous* | | | | | | | | |
| Original | −7.332 | 4.679 | 0.181 | 5.232 | −12.973 | 4.426 | 0.168 | 4.824 |
| Model I | −7.908 | 3.172 | −0.020 | 1.461 | −13.550 | 3.144 | −0.034 | 1.404 |
| Model II | −8.630 | 3.393 | 0.057 | 2.283 | −14.272 | 3.252 | 0.044 | 1.892 |
| *Herbaceous permanent— high productivity* | | | | | | | | |
| Original | −10.225 | 3.889 | 0.167 | 2.970 | −16.107 | 3.635 | 0.138 | 2.463 |
| Model I | −10.149 | 3.104 | −0.020 | 0.888 | −16.031 | 3.153 | −0.049 | 1.111 |
| Model II | −10.636 | 3.199 | 0.060 | 1.337 | −16.518 | 3.115 | 0.030 | 0.958 |
| *Herbaceous periodically* | | | | | | | | |
| Original | −8.753 | 3.310 | 0.152 | 1.022 | −15.354 | 3.223 | 0.130 | 0.814 |
| Model I | −8.622 | 3.131 | −0.024 | 0.686 | −15.223 | 3.101 | −0.046 | 0.490 |
| Model II | −8.785 | 3.187 | 0.065 | 0.632 | −15.386 | 3.107 | 0.043 | 0.306 |
| *Bushes and shrubs* | | | | | | | | |
| Original | −8.313 | 5.793 | 0.196 | 6.252 | −13.944 | 5.238 | 0.171 | 5.442 |
| Model I | −8.828 | 4.027 | −0.020 | 1.597 | −14.458 | 3.953 | −0.046 | 1.686 |
| Model II | −9.905 | 4.280 | 0.067 | 2.734 | −15.536 | 3.975 | 0.042 | 1.995 |
| *Bare rock and scree* | | | | | | | | |
| Original | −6.946 | 7.556 | 0.219 | 8.448 | −13.456 | 6.956 | 0.189 | 7.293 |
| Model I | −6.825 | 5.760 | −0.013 | 2.005 | −13.334 | 5.684 | −0.043 | 1.761 |
| Model II | −8.737 | 6.261 | 0.089 | 4.253 | −15.247 | 5.872 | 0.060 | 3.140 |

As suggested by the authors of [6], the backscatter dependency as a function of terrain aspect follows a sinusoidal type of behaviour and is quantified by its amplitude ($A$) modelled from a best fitting sine curve. The backscatter dependency as a function of slope steepness in range, instead, shows a linear type of behaviour and is quantified by the slope ($s$) of the best fitting linear function. In addition, the backscatter mean ($\mu$) and standard deviation ($\sigma$) are used to evaluate the performance of the models. A good model features both $A$ and $s$ close to zero, whereas the standard deviation $\sigma$ of the corrected backscatter should ideally correspond to the standard deviation within the same land cover class over flat terrain.

## 4. Results

The effect of the radiometric slope correction on the RGB Sentinel-1 composite is shown in Figure 4. The distinct topography of the area is noticeable in the original imagery (Figure 4a), with bright foreslopes towards west and darker backslopes towards east. After the applied slope correction with Model 1 (Figure 4b), the differences in backscatter rather reflect the various land cover classes (Figure 3b) instead of being influenced by the modulations along the slopes. The additional masking of layover and shadow affected areas according to Equations (7) and (8) is present on steep slopes foremost in the southwestern and northern part of the area of interest.

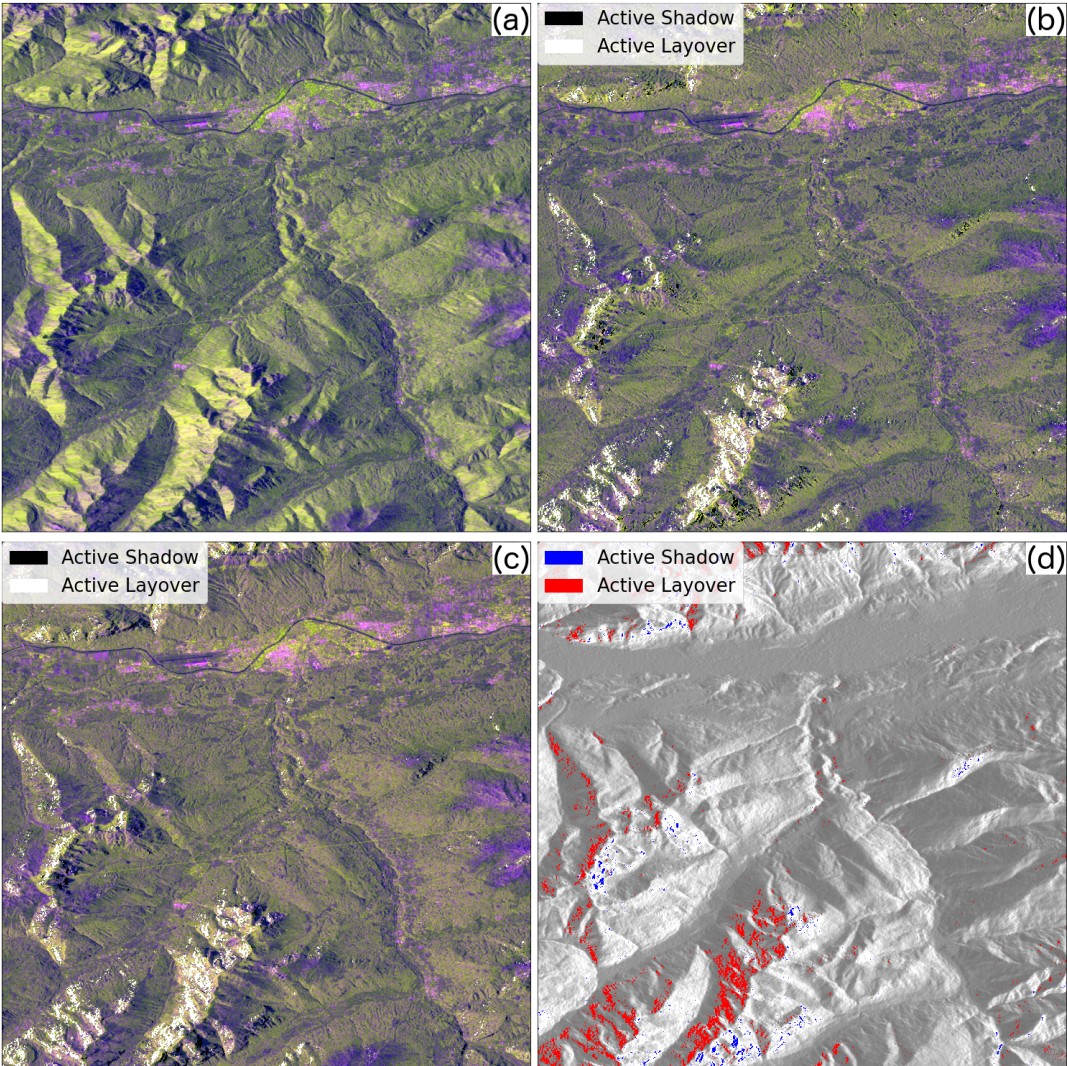

**Figure 4.** Sentinel-1 RGB color composite (R: $\sigma^0$-VV (dB), G: $\sigma^0$-VH (dB) B: VV/VH power ratio) over the Area of Interest before (**a**) and after correction with Model 1 (**b**) and Model 2 (**c**), as well as the difference of Model 1–Model 2 for the VV polarised bands stretched between −5 and 5 dB (**d**). Regions of active layover and shadow are overlaid in black and white (**b**,**c**) as well as in red and blue (**d**).

The Model 2 based correction shows similar results (Figure 4c), but residual effects are visible along the foreslopes oriented towards west where the topographic influence is still present. This is also reflected in the difference image (Figure 4d) of the VV-polarised bands from Model 1 and 2. Although being predominantly over vegetated areas, the discrepancy of both models on westward oriented foreslopes extends over all major land cover classes and amounts to as much as 3 dB. Differences of up to 5 dB appear as well on southwards oriented slopes. Being recognisable by more saturated green

tones in the corrected image, Model 1 seems to slightly overcorrect the radiometric distortions for slopes oriented in this direction.

The improvement of the corrections from both models with respect to their angular dependencies is shown graphically for the VV-polarized backscatter over coniferous trees in Figure 5. The data points of the image extend over the full range of values with regard to the two angles as depicted in the histograms of the y-axis. The strong angular dependence of the terrain-induced radiometric distortions on uncorrected SAR backscatter is visible in the top row of Figure 5. The uncorrected backscatter as a function of terrain aspect as shown in Figure 5a follows the typical sinusoidal type of behaviour, underlined by a high value of $A = 5.23$. Instead, Figure 5b shows the dependency between the uncorrected backscatter and the slope steepness in range, where backscatter increases linearly with increased slope steepness expressed by $s = 0.18$. The standard deviation $\sigma$ of the uncorrected backscatter amounts to 4.68 dB and is clearly influenced by the angular dependency.

The improvement of the corrections from both models is shown in the middle and bottom row of Figure 5. In both cases, the backscatter dependency is drastically reduced. For Model 1, the modulation of the backscatter with respect to the aspect angle still shows some residual dependency (Figure 5c), but is much closer distributed around the mean over the full range of aspect angles as compared to the original data (Figure 5a). This reflects in a drop of 3.8 dB in amplitude $A$ for the sinusoidal function fitted to the data. In line with the visual interpretation, southward-oriented slopes (i.e., $\phi_s \approx 180°$) are slightly above the mean and not fully corrected. With respect to the original data, southeastward-oriented slopes are overcorrected, while southwestern-oriented slopes are undercorrected. As shown in Figure 5e, the Model 2 based correction significantly reduces the dependency of the backscatter to the terrain's aspect as well, but performs slightly worse than Model 1 with regard to $A$. A residual sinusoidal behaviour is still visible, leading to undercorrected backscatter values towards east and southwest.

For the Model 1 based correction, the backscatter dependency with respect to the slope steepness angle $\alpha_r$ vanishes almost completely, shown by an equal distribution of all values around the mean (Figure 5d). The slope $s$ of $-0.02$ dB per degree slope steepness for the linear function fitted to the data underlines this. The slight negative trend is due to the apparent drop in backscatter for the slope steepness above 35°. Here, Model 1 is systematically overcorrecting, which results in unreliable low values along steep foreslopes. As this behaviour was described as well in [6], it is a characteristic of the model and not a sensor specific issue. Also the Model 2 correction shows a similar behaviour for slopes above 35° (Figure 5f). Nonetheless, the positive slope value of 0.06 reflects the residual influence on the corrected backscatter with regard to the full range of slope steepness angles in range, thus performing slightly worse than Model 1 with regard to the backscatter's dependency on slope steepness.

As illustrated in Table 1, the observations for the coniferous tree class are similar for all six land cover classes under investigation. The angular backscatter dependencies are generally reduced and both models improve the backscatter with respect to the original data. The offset of the backscatter mean between the models and the original data is generally within 1 dB. It should be noted that the observation of the mean depends on the distribution of the data points with respect to both angles. If for example more data points are present on foreslopes, the mean value will drop due to the correction. This is visible in the histograms of Figure 5b,d, where the mean values of the corrected image drops by 0.6 dB with respect to the original one, as there are more data points located on the positive slope steepness side.

The standard deviation is not affected by this issue and its interpretation is rather straightforward, meaning that a lower value generally indicates a better correction. The highest reductions for the Model 1 based correction are obtained for both forest types (between 1.2 and 1.5 dB). Herbaceous areas are most prominent over less inclined terrain and thus the reductions are rather small (0.2–0.5 dB). Model 2 does generally perform equal or worse than Model 1. While this has been observed as well in [6], it should nevertheless be considered that the selected classes are predominantly characterised by volume scattering.

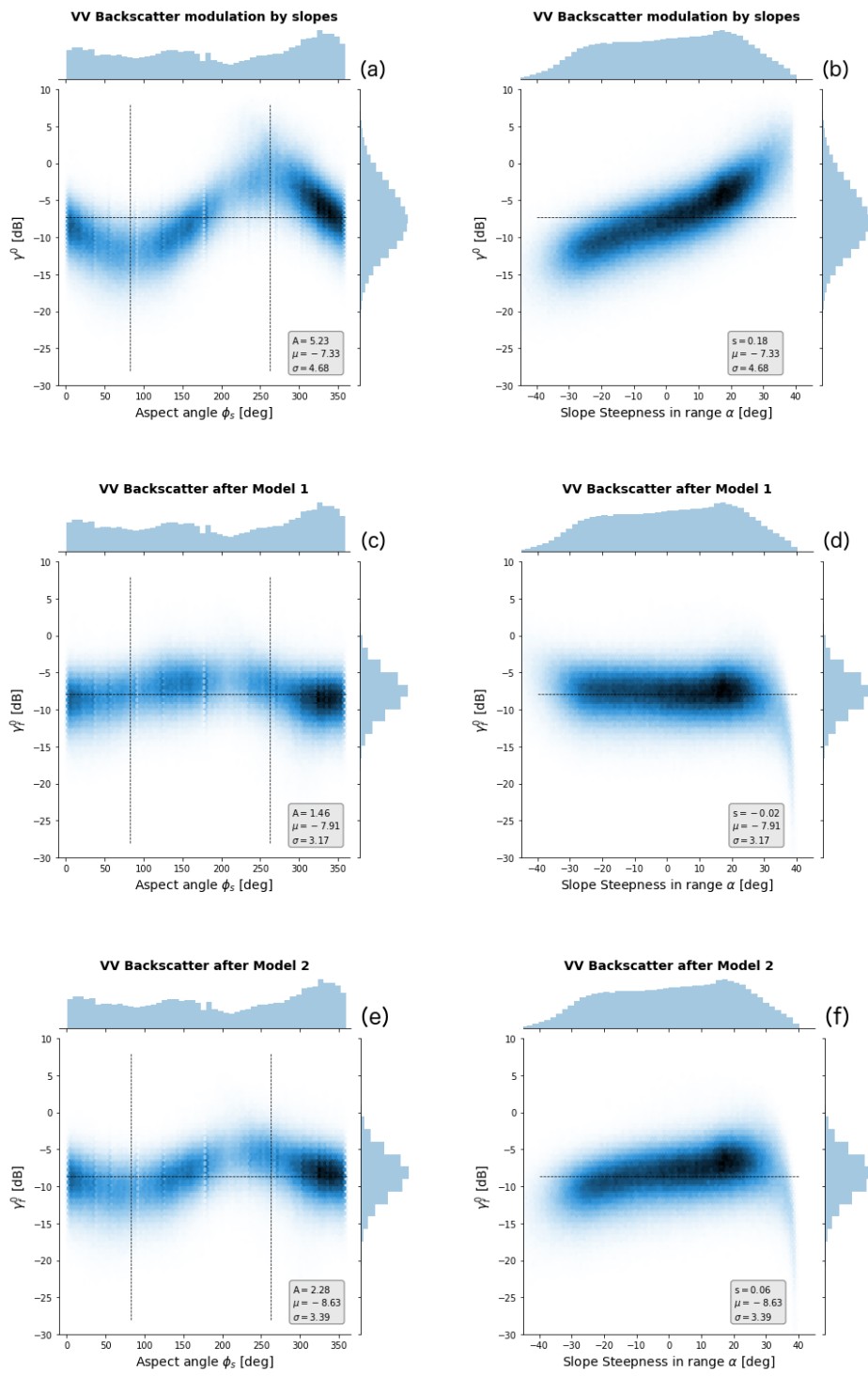

**Figure 5.** VV backscatter ($\gamma$) for coniferous trees as a function of aspect (left column) and slope steepness in range (right column) for: top row (**a**,**b**): original scene corrected for $\theta_i$, middle row (**c**,**d**): after correction for isotropic opaque volume scattering (Model 1), bottom row (**e**,**f**): after correction for isotropic surface scattering (Model 2). The vertical lines in the left column indicate the back- and forward-scattering directions. The horizontal lines indicate the mean backscatter level. The numbers in the bottom left stand for amplitude ($A$), slope ($s$), mean ($\mu$) and standard deviation ($\sigma$) of $\gamma$ in dB.

Moreover, with regard to the reduction of the backscatter's dependency on slope steepness, Model 1 performs generally better than Model 2. Model 1 shows a tiny residual decrease of backscatter with increasing steepness indicated by the slope of the fitted function. As discussed before, this is mainly due to a sharp drop of backscatter for terrain slopes with an inclination greater than 35°.

While the amplitude *A* of the fitted sine function of the backscatter with respect to the terrain aspect decreases significantly for both models over all land cover classes, none of the models is capable of completely removing its angular dependence with regard to terrain aspect. The Model 1 amplitude ranges from 0.7 to 2 dB for the VV and from 0.5 dB to 1.8 dB for the VH polarisation across all different land cover classes. Except for the periodical herbaceous land cover class, its performance is superior to the Model 2 based corrections with reduced amplitude values of up to 2.2 dB as compared to Model 2.

The masking of layover and shadow affected areas is shown in Figure 6. The major part of the layover affected areas are excluded by the active subregions in the unbuffered mask (Figure 6c). Only in the surroundings, bright backscatter values indicate the presence of passive subregions of layover. By increasing the buffer size (Figure 6d–f), those regions are getting masked out as well. However, as the simple consideration of a buffer does not take into account the actual geometrical complexity, some of the valid data points are getting excluded as well. This is especially visible for the shadow affected regions, where small areas are getting excessively extended with increased buffer size.

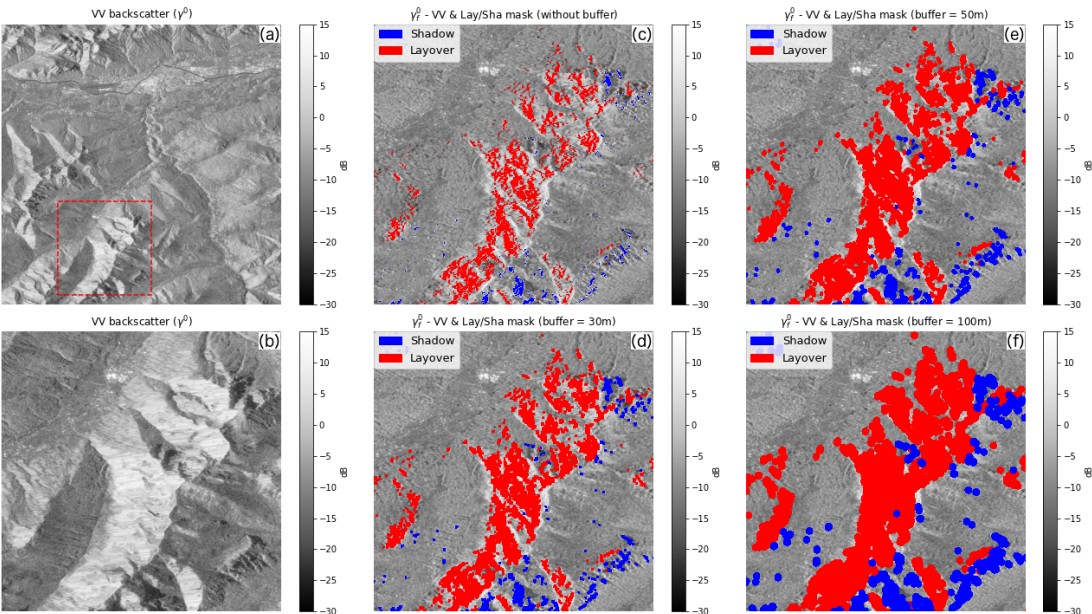

**Figure 6.** Comparison of the effect of the buffer parameter for the layover and shadow masks. Overview of study area (**a**) and zoom in (**b**) for VV backscatter ($\gamma^0$). Corrected VV backscatter ($\gamma^0_f$) with layover and shadow masks with no buffer (**c**), and added buffer size of 30, 50 and 100 metres, respectively, over the zoomed in area (**d–f**).

## 5. Discussion

### 5.1. Earth Engine Module for Slope Correction

A GEE module has been created to allow interested users the integration into their existent or new workflows (Supplementary Materials). The correction function takes a standard GEE Sentinel-1 image collection and returns a new image collection consisting of Sentinel-1 images with the corrected VV and VH bands as well as a no-data mask including both layover and shadow areas. Optional input parameters, given as a dictionary, allow (1) selecting the model, (2) the use of a DEM different than the default SRTM as well as (3) the buffer parameter for the no-data mask in meters. A detailed discussion on the selection criteria of each parameter for common usage is given below.

### 5.2. Model Selection

Although both physical reference models operate independently of terrain type, frequency and polarisation, their calculations are based on two different assumptions about the backscatter behaviour on ground. Model 1 assumes the backscatter taking place within a volume, thus being optimised for vegetated surfaces. Its application is therefore targeted on mapping of forest or crop parameters and has been successfully used in diverse studies [20–22]. Model 2 is optimised for surface scattering. Its application is in general advisable over urban areas as well as for studies of soil moisture or roughness over bare ground. For the presented case study, Model 1 shows a better overall performance, that has been also observed in former studies [6]. If the data should be used for land cover or land use studies, its usage seems favourable as compared to Model 2. However, this article demonstrates the improvement only exemplarily, and the actual choice should be information-based—meaning that a pre-assessment on the performance by the user itself is encouraged. This can be done visually, or in more depth, as presented within this article, by the available Jupyter notebooks that allow for the full reproduction of the shown statistics for any given area.

### 5.3. DEM Selection

The default DEM used for the slope correction and masking routine is the SRTM DEM with 30 meter pixel spacing [18]. It offers global coverage for all land masses between $-60°$ and $+60°$ latitude. Due to its free availability and quasi global coverage its usage is considered the common use case. As Sentinel-1 imagery features a 10 meter pixel spacing, its usage is actually suboptimal, as the terrain is not depicted at the same level of detail. If a higher resolution DEM is available for a given area, its usage should be considered. For areas outside the SRTM coverage, other medium to low resolution DEMs (e.g., ALOS World 3D) are available in GEE [23].

### 5.4. Layover & Shadow Mask

In line with the CEOS specifications [4], the application of the layover and shadow mask is strongly suggested, since the backscatter values over those areas are unreliable. The slope correction function does not mask the areas by default, so a dedicated operation needs to be undertaken for that. The additional use of the buffer parameter strongly depends on the area of interest and will be always determined by the trade-off between the coverage of passive subregions and affected pixels of valid data that might be masked out as well.

### 5.5. Drawbacks and Future Perspective

While drastically reducing the backscatter dependency with regard to the terrain geometry, both models show residual topographic effects with respect to the terrain orientation. In addition to inaccuracies of the DEM, those effects are linked to the assumption of homomorphism between map and radar geometry as well as the approximations of the incidence and azimuth angle for the image geometry. While it has been demonstrated that pixel-area-based slope correction methods, as the one described in [5], are more adequate to address this issue, their usage on the GEE platform for now is impractical. We therefore suggest the consideration of providing the pixel-area as an auxiliary band to each image. In this way, on-the-fly computations of pixel-area based slope corrections are feasible by overcoming the burden of computation between map and radar geometry. The same applies to the layover and shadow mask generation that should be ideally calculated in radar geometry before ingesting it to the platform. Therefore, both active and passive layover can be masked more adequately as compared to the simplified approach presented within this study.

## 6. Conclusions

This study demonstrates that angular-based radiometric slope corrections for Sentinel-1 imagery on the GEE platform are feasible by using the two established physical reference models

described. Indeed, the results are similar to those presented elsewhere with longer-wavelength SAR imagery [6] and the code is easily adaptable for other SAR missions that might be present on the GEE platform in future.

However, the proposed approach, based on simplified assumptions about the imaging geometry, does not fully compensate for the radiometric distortions as compared to more advanced methods [5,7,8]. As those methods rely on metadata that is not available on the platform, the use of the angular-based correction as presented within this study closes the gap between the discrepancy of the CEOS specifications for normalised backscatter ARD over land and the uncorrected data on the GEE platform on a best effort. Additionally, and in compliance with the voluntary standards on backscatter ARD of CEOS, a no-data mask is generated including areas affected by active layover and shadow. Therefore, the presented framework clearly overcomes important limitations and considerably improves the potential usage of Sentinel-1 imagery for a wide range of land applications such as land cover classification, deforestation monitoring, the retrieval of bio-geophysical parameters as well as the combination of imagery from different geometries.

**Supplementary Materials:** Reproducible Code S1 is available online at www.github.com/ESA-PhiLab/radiometric-slope-correction.

**Author Contributions:** The study set-up was done by all authors. Coding and processing as well as the preparation of the manuscript was done by A.V. A.M. revised the code and the manuscript and helped with the interpretation of the results. J.R. helped with the interpretation of the results and revised the manuscript. All authors have read and agreed to the published version of the manuscript.

**Acknowledgments:** The authors would like to thank Google for free access to the Google Earth Engine as well as the Google Earth Engine team for suggestions on optimizing the code. Contains modified Copernicus Sentinel data 2016.

**Funding:** The work of J.R and A.M. was partly funded through the US Government Silvercarbon programme.

**Conflicts of Interest:** The authors declare no conflict of interest.

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
