# Peer review of "Angular-Based Radiometric Slope Correction for Sentinel-1 on Google Earth Engine"

_remotesensing, doi:10.3390/rs12111867_

Round 1
Reviewer 1 Report
This research effort attempts to overcome an important challenge in SAR remote sensing; minimizing the impact of topographic effects on Sentinel-1 SAR backscatter values in complex terrains when using Google Earth Engine (GEE). With the ever increasing size of the GEE user community and the potential for use of GEE for the processing of high data volume Sentinel-1 collections, this effort seeks to provide solutions to what is likely the biggest technical challenges of using Sentinel-1 on this platform. Sentinel-1 is only available in a GRD format and the range-doppler radiometric terrain correction used in SNAP toolbox prior to GEE ingestion does not provide sufficient correction of terrain effects, especially slope-based effects. The authors attempt to overcome this issue by converting sigma0 imagery to gamma0 and then applying two terrain correction models, one assuming a volume scattering ground, and the other assuming a surface scattering ground. Further the authors provide a routine for layover and shadow masking. The justification for use of these models is present, and the performance of the correction models is well supported by the results in Table 1 which show clear reductions in slope based influence on backscatter values. The research effort is of clear value to the SAR and GEE communities, the study is carried out in a sound manner, and improvements of the terrain correction routines are clear.
I believe this manuscript is ready for publication pending the spelling and grammar corrections that I've noted below. Perhaps a re-evaluation of the transition between section 2 (methods) and section 3 (case study) could be performed. The description of the GEE Sentinel-1 processing in Section 3.1 after having described the terrain correction process in Section 2 does make things slightly confusing as the authors are applying their corrections to imagery that has been previously ingested into GEE. Describing the GEE ingestion process, then describing the application of terrain corrections would make more sense in terms of temporal ordering of processing steps. However, this is not a critical adjustment, just one that may enhance the organization of the manuscript.
Suggested edits by line number:
- “computationally intensive” rather than “compute intensive”
- “computationally intensive” rather than “compute intensive”
- “by a wider user community”
- “exceed” rather than “excel”
- artifacts
- May I suggest “SAR acquisition principles” or “SAR acquisition theory”.
- Should be “information” not “an information”
- “suited to areas” not “suited on areas”
63. “is based on” rather than “bases on” - It is based on the ratio…
Figure 2. In two cases the term “steeper then” was used. This should be changed to “steeper than”
101. Might be a further distance to
128. Acquired on 15.08.2016
After re-reading this paper, it seems that organizationally, there may need to be some slight modifications between section 2 and section 3 to communicate more sequential processing steps. The end of section 3.1 could include a few sentences about how the Model-1 and Model-2 corrections were applied to imagery within GEE. The descriptions and derivations of these models in section 2 was adequately described, but describing the GEE implementation of the models would provide clarify to the reader. It’s somewhat confusing that section 3.1 ends with a description of how Sentinel-1 imagery is processed in SNAP before ingestion in GEE and then the next section simply talks about evaluation of the author’s terrain corrections. The application of Model-1 and Model-2 in GEE are missing steps in section 3. Alternatively, the discussion of the SNAP-GEE processing could be moved to the beginning of section 2.1 before the discussion of Model-1 and Model-2 derivations. Either of these changes would improve the clarity of the manuscript.
151. It needs to be clarified what figure this is referring to.
154. This needs to reference figure 3.
172. How is the masking here being performed? If this masking is an implementation of equations 5 and 6, this needs to be stated explicitly.
176. This is also reflected in the difference image.
201 “is generally within”
211. “worse than model-1”
215. “than” not “then”
Figure 6. “Comparison of” rather than “comparison on”
242. “then” not “than”
246. “are based on” rather than “base on”
248. Remove the term “bio-geophyiscal”
Author Response
Dear reviewer,
many thanks for the constructive feedback to our manuscript.
Below find the adaptions we've undertaken.
- “computationally intensive” rather than “compute intensive” adapted
- “computationally intensive” rather than “compute intensive”adapted
- “by a wider user community” adapted
- “exceed” rather than “excel” adapted
- artifacts adapted
- May I suggest “SAR acquisition principles” or “SAR acquisition theory”. adapted
- Should be “information” not “an information” adapted
- “suited to areas” not “suited on areas” adapted
63. “is based on” rather than “bases on” adapted - It is based on the ratio… adapted
Figure 2. In two cases the term “steeper then” was used. This should be changed to “steeper than” adapted
101. Might be a further distance to adapted
128. Acquired on 15.08.2016 adapted
After re-reading this paper, it seems that organizationally, there may need to be some slight modifications between section 2 and section 3 to communicate more sequential processing steps. The end of section 3.1 could include a few sentences about how the Model-1 and Model-2 corrections were applied to imagery within GEE. The descriptions and derivations of these models in section 2 was adequately described, but describing the GEE implementation of the models would provide clarify to the reader. It’s somewhat confusing that section 3.1 ends with a description of how Sentinel-1 imagery is processed in SNAP before ingestion in GEE and then the next section simply talks about evaluation of the author’s terrain corrections. The application of Model-1 and Model-2 in GEE are missing steps in section 3. Alternatively, the discussion of the SNAP-GEE processing could be moved to the beginning of section 2.1 before the discussion of Model-1 and Model-2 derivations. Either of these changes would improve the clarity of the manuscript. We adapted this changes, but in a slight different fashion. There is a new section 2.1 that explains the ingestion process done by Earth Engine. Regarding an extra section on how it is implmented in Eath engine would have led to a lot of redundancy to section 2.2 as well as to the Appendix A. During the elaboration of this revision we found that combing your request of adding info on the specifics of EE implementation with the theory is the less redundant option. We hope that satisifies your request
151. It needs to be clarified what figure this is referring to. For the evaluation scheme we avoided to link extensively to the shown figures, since it should rather describe the general theory behind the evaluation of the improvements. However, also on request of another reviewer, we added more references to the shown figures in the result part
154. This needs to reference figure 3. added
172. How is the masking here being performed? If this masking is an implementation of equations 5 and 6, this needs to be stated explicitly. added
176. This is also reflected in the difference image. adapted
201 “is generally within” adapted
211. “worse than model-1” adapted
215. “than” not “then” adapted
Figure 6. “Comparison of” rather than “comparison on” adapted
242. “then” not “than” adapted
246. “are based on” rather than “base on” adapted
248. Remove the term “bio-geophyiscal” adapted
Reviewer 2 Report
The manuscript describes an experiment testing software to perform radiometric terrain correction on Sentinel-1 imagery in the Google Earth Engine (GEE).
As the authors briefly introduce, the GEE provides sigma0 backscatter values generated under the assumption of a flat Earth.
Their algorithm is meant to correct for that flat Earth assumption, performing DEM-based corrections to the backscatter values.
The authors lean on a self-cited RSE paper [6] with a common co-author, but ignore some critical deficiencies in that paper's approach.
Critically, working only in map geometry ignores differences between slant range geometry of the radar measurements and the map geometry of the GEE.
The authors obliquely mention on p. 2 line 55 that other correction methods require state vectors and cannot operate directly in the GEE map geometry, but then appear to want to claim that deficiency as an *advantage* rather than a failure of the method.
In fact, by ignoring all the losses that go with an assumption of homomorphism, the slope corrections applied fail to account for basic properties of radar image acquisition, i.e. the many-to-one nature of connections between map and slant range geometry and the *integration* inherent in radar acquisitions of foreslopes. By quietly (this is unmentioned) assuming that the slant range and map geometries are homomorphic, their results lose traction with real data. Their assumption of homomorphism implies that they are unable to capture the effects of passive layover and shadow. In both cases, the passive part is often larger than the active, so most of the corrections possible using a DEM are never generated or applied.
Rather as on p. 2 line 64 stating that it is "straightforward" to perform radiometric slope correction based on "angular relationships", one should directly use local estimates of "contributing area", as shown in [5]. That publication showed that if one chooses to use angles as proxies for areas under a homomorphic assumption, the corrections are doomed to failure.
More useful than the current manuscript would be a comparison of the corrections made using [5] (multiple implementations exist) vs. [6].
Slope estimates based on neighbouring map geometry pixels cannot be equivalent to those done given actual knowledge of the *integrative* nature of summation of DEM-portions into a set of slant range geometry "buckets". Methods able to go beyond a local neighborhood will produce better corrections. They pay a price with a need for extra processing but provide higher quality. They cannot assume that use of local angles is enough to explain terrain-induced radiometric distortions.
Detailed comments
1. Equation (1)
This equation is only true for ellipsoidal inc. angles, not on terrain
2. Equations (2) and (3) both choose to use angles as proxies for area, rather than area itself directly, which can be calculated given a DEM.
3. Figure 2 and Equations (4) and (5)
In the label "The red line depicts active layover areas that can be derived from the angular dependencies"
Here the authors misunderstand the nature of layover. They should refer to a standard text for the definition of layover,
e.g. Meier E., Frei U., Nuesch D., Precise terrain corrected geocoded images, SAR Geocoding: Data and Systems, G. Schreier, Wichmann, 1993.
Layover cannot be determined by any single local angle. It arises in a neighborhood from the relation between a point under consideration
in relation to other points, possibly quite a distance away in ground and slant range.
Using "angular dependencies" makes a calculation simple, but it ignores how it is that layover actually arises in the real world.
As a consequence, large swaths of sections 3.2 and 3.3 lose impact, due to use of models that are easy to calculate but wrong in the real world.
4. Figure 4
No radiometric legend is provided. Please declare the black saturation and white saturation dB values for R, G, and B.
The authors currently do not explain the cause of the large white regions in the SW of (b) and (c ).
No clear separate descriptions are provided for images (a) through (d)
In the text, the authors state that they only map "active" layover and shadow, but then in the figures (e.g. Figure 4) the label indicates simply "Layover" and "Shadow", with no limitation.
5. Figure 5
It is unclear why the authors chose only a small subset of coniferous pixels (never indicated on a map) for this figure.
Was the performance significantly different for other terrain types? It would be of benefit to be able to indicate general applicability of a method, without resort to "cherry picking" results.
6. Table 1
This large set of numbers for a small single image invites the question of whether or not overfitting using many degrees of freedom is unrealistically "improving" the results.
7. Figure 6
The order from a, c, d, b, e, f is unnecessarily ambiguous and not intuitive.
It is unclear why users should wish to use such a "buffer" solution in place of better corrections that would be available using methods that operate directly in radar geometry.
This is a point currently missing in the manuscript, though it would deserve strong prominence in setting limits to the algorithm's potential market.
8. p. 10 line 242
"The model that should be used"
Would not using a model that operates in radar geometry be preferable?
If that is not possible in GEE, that should be mentioned prominently in the introduction and conclusions to avoid giving readers false hope for the availability of best available quality results.
9. p. 10 line 253
"If the data should be used for land cover or land use studies, its usage seems favourable"
This favourability should be demonstrated on a quantitative basis in comparison to competing algorithms.
10. Section 4.3
The GEE pre-processing as currently done as well as future plans should be discussed.
It should be made clear that this pre-processing is currently *not* user selectable.
11. Conclusion
"This study demonstrates that the radiometric slope correction..."
It is unclear how the definite article "the" was earned in this phrase. There are many possible different corrections that could be applied. In this manuscript, the authors do not compare their chosen method with any others, even though multiple implementations of others exist. The manuscript would strongly benefit from a comparison of their method against others that actually map regions affected also by "passive layover and shadow" and that account for differences between radar and map geometry.
Typos e.g.:
P. 6 line 155 "build-up" --> "built up"
P. 9, line 220 "exemplary shown" --> "exemplarily shown"
Author Response
The manuscript describes an experiment testing software to perform radiometric terrain correction on Sentinel-1 imagery in the Google Earth Engine (GEE).
As the authors briefly introduce, the GEE provides sigma0 backscatter values generated under the assumption of a flat Earth.
Their algorithm is meant to correct for that flat Earth assumption, performing DEM-based corrections to the backscatter values.
The authors lean on a self-cited RSE paper [6] with a common co-author, but ignore some critical deficiencies in that paper's approach.
Critically, working only in map geometry ignores differences between slant range geometry of the radar measurements and the map geometry of the GEE.
The authors obliquely mention on p. 2 line 55 that other correction methods require state vectors and cannot operate directly in the GEE map geometry, but then appear to want to claim that deficiency as an *advantage* rather than a failure of the method.
In fact, by ignoring all the losses that go with an assumption of homomorphism, the slope corrections applied fail to account for basic properties of radar image acquisition, i.e. the many-to-one nature of connections between map and slant range geometry and the *integration* inherent in radar acquisitions of foreslopes. By quietly (this is unmentioned) assuming that the slant range and map geometries are homomorphic, their results lose traction with real data. Their assumption of homomorphism implies that they are unable to capture the effects of passive layover and shadow. In both cases, the passive part is often larger than the active, so most of the corrections possible using a DEM are never generated or applied.
Rather as on p. 2 line 64 stating that it is "straightforward" to perform radiometric slope correction based on "angular relationships", one should directly use local estimates of "contributing area", as shown in [5]. That publication showed that if one chooses to use angles as proxies for areas under a homomorphic assumption, the corrections are doomed to failure.
More useful than the current manuscript would be a comparison of the corrections made using [5] (multiple implementations exist) vs. [6].
Slope estimates based on neighbouring map geometry pixels cannot be equivalent to those done given actual knowledge of the *integrative* nature of summation of DEM-portions into a set of slant range geometry "buckets". Methods able to go beyond a local neighborhood will produce better corrections. They pay a price with a need for extra processing but provide higher quality. They cannot assume that use of local angles is enough to explain terrain-induced radiometric distortions.
Dear reviewer,
we appreciate your critics and assure that we are aware of the limitations with respect to other methods, specifically with respect to the one of Small 2011. We fully acknowledge that pointing to the deficits of the used approach needed to be improved, and we hopefully accomplished this in the revised version of the manuscript. We now also make it clear right from the start that we deal with a angular based approach by having changed the title of the paper and abstract accordingly (note, not visible in the diff file because of issues of latexdiff program). In addition, we added a dedicated subsection into the discussion.
- Abstract addition:
“Furthermore, suggestions for specific use cases are discussed and drawbacks of the method with respect to pixel-area based methods are highlighted.”
- addition to Introduction, p.2 line 59-69:
“As with all angular-based approaches, both models feature the drawback of not accounting for the apparent heteromorphic relation between map and radar geometry as discussed in Small2011. Both Small2011 and Frey2013 show that pixel-area based approaches for the radiometric slope correction are superior by considering the actual topological relationships between both geometries and thus take all the underlying basic properties of the radar image acquisition into account. However, the back- and forward computation between map and radar geometry of each image would not only heavily affect performance, but also requires the availability of the state orbit vectors (i.e.~the exact position of the satellite during the acquisition), an information that is left out during the ingestion process on GEE. As a result, the selection of an adequate correction procedure on the GEE platform is limited and the use of an angular-based approach that is based on simplified assumptions remains the only feasible option under the current preconditions.”
- and discussion 5.5, where we added a new subsection
“While drastically reducing the backscatter dependency with regard to the terrain geometry, both models show residual topographic effects with respect to the terrain orientation. In addition to inaccuracies of the DEM, those effects are linked to the assumption of homomorphism between map and radar geometry as well as the approximations of the incidence and azimuth angle for the image geometry. While it has been demonstrated that pixel-area based slope correction methods, as the one described by Small2011}, are more adequate to address this issue, their usage on the GEE platform for now is impractical.”
Further, we would like to provide some clarifications, since most of your comments aim at other methods in radar geometry to which the method proposed should be compared. It is not the scope of the article to compare it to those methods, because of the fact that the ones based on radar geometry are impractical on GEE (at least for now) and we do specifically address GEE users (as stated in the title). From our point of view a comparison wouldn’t make sense if a completely different software set-up needs to be used. In addition, comparisons have been already published. We added the paper of Frey 2013 and mention it now explicitly in the introduction, acknowledging that pixel-area based approaches are more adequate and accurate.
Ultimately, we took our initial selection from the CEOS suggestions on ARD standards, and ended up with the only possible implementation on the platform. For this our implementation should be considered as a best effort for the specific preconditions. We added this specifically in conclusion:
conclusion p.14 line 357-362.
“However, the proposed approach, based on simplified assumptions about the imaging geometry, does not fully compensate for the radiometric distortions as compared to more advanced methods. Since those methods rely on metadata that is not available on the platform, the use of the angular-based correction as presented within this study closes the gap between the discrepancy of the CEOS specifications for normalised backscatter ARD over land and the uncorrected data on the GEE platform on a best effort.”
We are still convinced that our manuscript is of high relevance. In 6 years of availability of Sentinel-1 data, and even more for Google Earth Engine, there does not seem to be a solution to this issue for that specific platform. Furthermore, users of GEE are mainly addressing large-scale problems where computational performance is of higher relevance than quality. In addition, we see a high demand on the use of SAR data on the platform from the user forum that goes far beyond the traditional SAR expert community. Those readers are the “target group” of this article, i.e. giving non SAR experts a possibility to use “better” SAR data as currently available.
As with all models the angular-based approach relies on simplifications and certain assumptions. This is even true for the approach of Small 2011, since the underlying terrain geometry might be inaccurate for various reasons. However, that does not mean that they are not useful.
Since we are aware that pixel-area based approaches are more accurate, we updated the manuscript with suggestions not only to Earth Engine, but rather with regard to the SNAP software (that is used by the GEE team to pre-process the data) and the ingestion process:
- Discussion 5.5, p.13 line 346-350
We therefore suggest the consideration of providing the pixel-area as an auxiliary band to each image. In this way, on-the-fly computations of pixel-area based slope corrections are feasible by overcoming the burden of computation between map and radar geometry. The same applies to the layover and shadow mask generation that should be ideally calculated in radar geometry before ingesting it to the platform. Hence, both active and passive layover can be masked more adequately.”
Even if the state-orbits would be included into S1’s metadata of Earth Engine, addressing the problem of heteromorphity would be computationally very costly (as also mentioned within your review). The advantage of Earth Engine showing results almost instantly would vanish. We think the best approach is to have an option in SNAP to export the pixel area, and add it as a geocoded band, so that on-the-fly calculations would then be possible on GEE, leaving the user still the option to choose and to come up with comparisons. Until this happens we think the adaption of the approach by Hoekman et al is one of the most practical given the limitations of GEE and hope under this perspective you can give your ok for the publication of this manuscript.
Many thanks,
The authors
Detailed comments
1. Equation (1)
This equation is only true for ellipsoidal inc. angles, not on terrain
Indeed, the IA as provided in earth engine is from the viewing perspective, and not corrected for the ellipsoid. This simplification is usually done for aerial SAR, but should be addressed for spaceborne SAR. The problem is the missing position of the sensor, for which the off-nadir angle cannot be computed. We added a phrase on p. 4 line 139 that states that the estimation of gamma0 is in this case only an approximation.
“It should be noted that the incidence angle $\theta_{i}$ on GEE is given as the viewing incidence angle, therefore neglecting the earth curvature's influence on $\theta_{i}$ on the ground. Thus, the resulting $\gamma^0$ of equation 4 represents only an approximated estimate.”
Surely it affects the routine, and we add this to the new subsection 5.5:
“While drastically reducing the backscatter dependency with regard to the terrain geometry, both models show residual topographic effects with respect to the terrain orientation. In addition to inaccuracies of the DEM, those effects are linked to the assumption of homomorphism between map and radar geometry as well as the approximations of the incidence and azimuth angle for the image geometry.”
2. Equations (2) and (3) both choose to use angles as proxies for area, rather than area itself directly, which can be calculated given a DEM.
As stated above, we use a simplified assumption model because of the limitations of GEE
3. Figure 2 and Equations (4) and (5)
In the label "The red line depicts active layover areas that can be derived from the angular dependencies"
Here the authors misunderstand the nature of layover. They should refer to a standard text for the definition of layover,
e.g. Meier E., Frei U., Nuesch D., Precise terrain corrected geocoded images, SAR Geocoding: Data and Systems, G. Schreier, Wichmann, 1993.
Layover cannot be determined by any single local angle. It arises in a neighborhood from the relation between a point under consideration
in relation to other points, possibly quite a distance away in ground and slant range.
Using "angular dependencies" makes a calculation simple, but it ignores how it is that layover actually arises in the real world.
As a consequence, large swaths of sections 3.2 and 3.3 lose impact, due to use of models that are easy to calculate but wrong in the real world.
Again, because of the limitations of GEE, we use a simplified model that has been published elsewhere and was successfully applied in cited studies. It should be noted that the aspect angle and slope angle are also computed based on neighbourhood. Since both are the basis for the angular based calculation, at least for the active part, those considerations are not identical, but close to reality. We acknowledge that in certain circumstances it will fail to reproduce real world results because of the lack of consideration of further neighbourhood. We mention this now in the introduction and Section 2.3.
We also admit that the buffer for the passive part, generally, is not an ideal solution, but it is tailored to the platform, where sensor position is not available. In addition the user is warned in the Discussion section and it is an optional parameter that is not necessary to use.
4. Figure 4
No radiometric legend is provided. Please declare the black saturation and white saturation dB values for R, G, and B.
The authors currently do not explain the cause of the large white regions in the SW of (b) and (c ).
No clear separate descriptions are provided for images (a) through (d)
In the text, the authors state that they only map "active" layover and shadow, but then in the figures (e.g. Figure 4) the label indicates simply "Layover" and "Shadow", with no limitation.
The labels have been changed and made bigger to be clear that we consider active layover and shadow, and it has been added to the capture as well.
For the greyscale images in Figure 6 we have a radiometric legend, but for the RGB it is impossible to have a radiometric legend.
As shown within the figures, the black and white regions refer to active layover and shadow. Since our images are coloured, we found this the most appropriate “colours” one can choose to overlay this information. Otherwise misinterpretation between the mask and the data is more likely. Greyscale images have been considered, but it is much more difficult to interpret the LC differences that are important for interpretation as well. In this context it should also be noted that this figure is rather intended to visually interpret the improvements. For the layover and shadow we have Figure 6, where a radiometric legend is already there, and LS areas are far more visible, also because of the zoom in. In addition, the coloured LS mask in Figure 4d should help to identify these areas in Figure 4b and c
5. Figure 5
It is unclear why the authors chose only a small subset of coniferous pixels (never indicated on a map) for this figure.
Was the performance significantly different for other terrain types? It would be of benefit to be able to indicate general applicability of a method, without resort to "cherry picking" results.
We selected coniferous forest, because it is the most prominent class that covers the full range of steepness and aspect. The number of pixels within this class is also considerable. Overall we used 3000x3000 pixels, and at least 10 % are coniferous forest.
In the Jupyter notebooks it is possible to fully re-create the plots for all classes. Showing them all would affect the readability of the article. All of them show similar results, as summed up in the Table 1 and visible in the corrected RGB images. Also note that the selection of the 6 classes shown is based on the distribution of the pixels with respect to the terrain angles. E.g. urban areas are foremost on flat terrain so that pixels over areas of steeper terrain are missing to accurately fit the evaluation functions. Therefore they have been removed.
6. Table 1
This large set of numbers for a small single image invites the question of whether or not overfitting using many degrees of freedom is unrealistically "improving" the results.
Not 100% sure what you mean by this large set of numbers. Regarding the model we use a physical model where overfitting is not possible. Nothing is fitted to the data. In addition, it is a case study to exemplarily show the improvements. We mention in the discussion that we do not assume general behavior in terms of performance of the 2 models and encourage the reader to evaluate themselves case by case.
If you mean the evaluation parameters, which are fitted, it is clear from the uncorrected data that the angular dependency follows this type of functions. Variation in the backscatter is expected within a single class because of speckle, but also errors in the underlying land cover classification that represents a simplified view on the real world itself. We also tried to describe the improvements in relative terms, e.g. Amplitude is going down by 3.8 dB and do not insist that the actual numbers are representative.
7. Figure 6
The order from a, c, d, b, e, f is unnecessarily ambiguous and not intuitive.
It is unclear why users should wish to use such a "buffer" solution in place of better corrections that would be available using methods that operate directly in radar geometry.
This is a point currently missing in the manuscript, though it would deserve strong prominence in setting limits to the algorithm's potential market.
Because of the already mentioned limitations of GEE and the simplification of our LS masking approach we provide a fast solution to give the user the possibility to mask out irrelevant data. As stated in the article this comes with the cost of likely removing valid areas as well. We know that this is not a very appropriate way of doing, but because of the lack of metadata is on a best effort. In addition, it is optional, so that a user can decide on his own if this makes sense or not for his specific use case.
The letters should indicate in which way the figure should be read. Try to imagine the image with a,b,c,d,e,f. In our opinion this leads to more confusion.
8. p. 10 line 242
"The model that should be used"
Would not using a model that operates in radar geometry be preferable?
If that is not possible in GEE, that should be mentioned prominently in the introduction and conclusions to avoid giving readers false hope for the availability of best available quality results.
Yes, we would very much like to see a more precise pixel-area based implementation in GEE, but because of the limitations we are limited to angular based approach. We mention this now clearly in the introduction, as well as in the conclusion and provide recommendations how this could be achieved. But this needs considerable effort by the operator of the platform. For this reason we consider our approach a valid alternative for now. We give suggestions in the conclusions how such an approach could be realised.
p.13 line 350
“We therefore suggest the consideration of providing the pixel-area as an auxiliary band to each image. In this way, on-the-fly computations of pixel-area based slope corrections are feasible by overcoming the burden of computation between map and radar geometry. The same applies to the layover and shadow mask generation that should be ideally calculated in radar geometry before ingesting it to the platform. Hence, both active and passive layover can be masked more adequately.
9. p. 10 line 253
"If the data should be used for land cover or land use studies, its usage seems favourable"
This favourability should be demonstrated on a quantitative basis in comparison to competing algorithms.
Out intention with this phrase was to show that Model 1 should be favoured over Model 2. You are right that it is not clear that there might be other correction procedures that are better suited for LCLU studies. We rephrased the sentence to be more clear.
“If the data should be used for land cover or land use studies, its usage seems favourable as compared to Model 2”
10. Section 4.3
The GEE pre-processing as currently done as well as future plans should be discussed.
It should be made clear that this pre-processing is currently *not* user selectable.
This is included into the new subsection 5.5 with added recommendations for the future. Having a user-selected pre-processing would however make the advantages of ARD data, as mentioned in the very first paragraph of the paper, obsolete. We do not think EE should switch to user defined pre-processing, but rather think about what auxiliary information they want to add to the ingestion process. If a proper LS mask and pixel area would be added to the imagery as auxiliary bands, on the fly calculations, as typical for GEE, would still be possible, and user have a freedom to choose which method, if any, they want to use. We admit that if this becomes a reality, our method will likely be obsolete, but it does not seem to be the case within the near future.
11. Conclusion
"This study demonstrates that the radiometric slope correction..."
It is unclear how the definite article "the" was earned in this phrase. There are many possible different corrections that could be applied. In this manuscript, the authors do not compare their chosen method with any others, even though multiple implementations of others exist. The manuscript would strongly benefit from a comparison of their method against others that actually map regions affected also by "passive layover and shadow" and that account for differences between radar and map geometry.
You are right, the formulation is misleading and has been changed. to:
“This study demonstrates that angular-based radiometric slope corrections ...“
As said before, it is not the aim of this article to compare with other methods which are impractical to implement in GEE, since the method is fully targeted on GEE users. We followed the CEOS suggestions where the 3 most prominent methods are outlined. As already stated in the introduction we go for the one in map geometry because of the apparent limitations.
Typos e.g.:
P. 6 line 155 "build-up" --> "built up" corrected
P. 9, line 220 "exemplary shown" --> "exemplarily shown" corrected
Reviewer 3 Report
The paper by Andreas Vollrath et al is a carefully written contribution. It provides a radiometric slope correction routine for Sentinel-1 SAR imagery on the Google Earth Engine platform implementing two former established physical reference models. I recommend accepting it for publication in Remote Sensing after minor revision from the authors.
- Capture for Figure 4: Reference to (d) is lacking.
- Capture for Figure 5: Not one of references (a) to (d) has been used in the capture.
- Not all of the references (a) to (d) of the Figure 5 have been explained or referred in the text.
- For the numbers presented in the bottom left of Figure 5, amplitude (A), slope (s), mean (µ) and standard deviation (σ) of γ, it could be beneficial for many readers if these parameters would also explained schematically and shown for example in Figure 5 (a) and (b).
- I think, Figure 7 and Appendix B could be omitted. Instead, the bottom row of Figure 7 could be added to the Figure 5 as parts (e) and (f).
- Modification of Figure 5 would be helpful also for reviewing data presented in Table 1.
Author Response
Dear reviewer,
many thanks for the constructive feedback for our manuscript.
See the following changes:
- Capture for Figure 4: Reference to (d) is lacking. added
- Capture for Figure 5: Not one of references (a) to (d) has been used in the capture.added
- Not all of the references (a) to (d) of the Figure 5 have been explained or referred in the text.added
- For the numbers presented in the bottom left of Figure 5, amplitude (A), slope (s), mean (µ) and standard deviation (σ) of γ, it could be beneficial for many readers if these parameters would also explained schematically and shown for example in Figure 5 (a) and (b). from our point of view putting schematics in figure 5 might overload the figure and lead to confusion. Instead, we added some interpretation of the top row to the results section so that the readers can follow the not so directly intuitive concept of evaluation there and see on which basis we evaluate the improvements. I hope this is an acceptable compromise.
- I think, Figure 7 and Appendix B could be omitted. Instead, the bottom row of Figure 7 could be added to the Figure 5 as parts (e) and (f). very valuable comment, we did so
- Modification of Figure 5 would be helpful also for reviewing data presented in Table 1. exactly